# The Role of a Restorative Resource in the Academic Context in Improving Intrinsic and Extrinsic Motivation and Flow within the Job Demands–Resources Model

**DOI:** 10.3390/ijerph192215263

**Published:** 2022-11-18

**Authors:** Diego Bellini, Barbara Barbieri, Massimiliano Barattucci, Maria Lidia Mascia, Tiziana Ramaci

**Affiliations:** 1Faculty of Medicine, University of Cagliari, 09042 Cagliari, Italy; 2Department of Political and Social Sciences, University of Cagliari, 09123 Cagliari, Italy; 3Department of Human and Social Sciences, University of Bergamo, 24129 Bergamo, Italy; 4Department of Pedagogy, Psychology, Philosophy, University of Cagliari, 09123 Cagliari, Italy; 5Faculty of Human and Social Sciences, Kore University of Enna, 94100 Enna, Italy

**Keywords:** motivation, restorative environments, stress, academic context, flow

## Abstract

The perceived quality of the learning environment may influence both motivation and concentration. Little is known about how perceived characteristics of the learning environment, and specifically sub-dimensions of Perceived Restorativeness (being away, fascination, compatibility, and extent), can promote these positive effects in an academic context. We addressed, through a correlational study, the possibility that the characteristics of learning environments may promote concentration and involvement in activity (i.e., flow) via intrinsic and extrinsic motivation for academic study within the job demands–resources model. A total of 165 Italian university psychology classes in a 3-year degree course from two different universities context completed an online questionnaire made up of the construct considered in this study. Results in the hierarchical multivariate regression analyses confirm that the restorative quality of learning environments (i.e., being away, compatibility, extent) is positively correlated with flow. However, there is a non-significant relationship between extent and flow. Regression analyses show a significant indirect effect of compatibility, both through intrinsic and extrinsic student motivation. Furthermore, the results confirm a significant indirect effect of extent through intrinsic motivation and being away, and fascination through Extrinsic motivation. Furthermore, intrinsic motivation is a full mediator between the extent and flow relationship. The results underline the importance of considering the restorative quality of the environment for improving place design, concentration, and student learning motivation.

## 1. Introduction

Motivation is an essential element of learning and a determinant of academic success [1,2]. Notably, high levels of student motivation lead to higher levels of productivity [3]. Therefore, university students need to improve their motivation levels to increase academic achievement [4].

Research has shown that individuals who are highly motivated can experience an intrinsically enjoyable mental state [5,6]. This positive state of involvement and absorption in a task was defined by Csikszentmihalyi [7,8] as flow. Specifically, people who experience the state of flow are fully concentrated and engaged, wholly absorbed in their activity [9], and show high performance in learning [10]. Furthermore, perceptions related to the learning environment (e.g., light conditions, noise, and features of the environments) may affect motivation [11] and performance [12,13,14]. Authors have noted that environmental stressors in the learning context may reduce motivation and increase learned helplessness [15]. When students deal with academic demands such as environmental stressors (i.e., aspects that require sustained physical or mental effort), they deplete resources (aspects in the job or learning environment that are functional in achieving goals) such as motivation and direct attention, which can reduce performance [16]. Resource depletion leads to the need to restore them [17]. Students may benefit from a restoration process that renews the psychological resources of depleted attention [18].

It is therefore important to take into account which physical and social features of learning environments are potentially significant in restoring individual resources and increasing their motivation and flow [19]. To examine the role of the physical and social features of the learning environment, we used the job demands–resources model (JD-R) [20,21] and the attention restoration theory (ART) [22,23].

The JD-R model is one of the most used frameworks for examining the relationship between resources and demands in the work and learning context. Specifically, this model describes the interaction between job characteristics, student motivation and engagement [24,25], and positive outcomes such as improved student performance [26,27].

More specifically, the job demands–resources model describes how job resources (such as characteristics of the learning environment) can promote positive outcomes (such as flow) through a motivational process or reduce the negative effect of job demands through job and personal resources [28].

The JD-R model is also applied to both the positive consequences of studying [25,26] and the negative consequences of studying [29,30]. However, to date, the model has yet to receive sufficient attention within the university context.

Furthermore, although the model has been widely used and has considered a variety of job characteristics able to promote positive outcomes, the role of physical and social features as environmental resources has yet to receive attention. Observing that there is little research on this topic, authors have recently put forward several studies describing the positive effect of the environment, i.e., restorativeness, on positive outcomes such as job satisfaction, work engagement [11], fatigue [31], and organizational behaviors [32]. Notably, these authors examined the role of environmental restorative resources in the work context. However, the role of the restorative characteristics of the environment on motivation and flow has not yet been considered or studied.

Specifically, the learning environment, such as the university campus or home, may serve as a promoting resource for motivation and flow state.

We investigate how characteristics of a learning environment directly promote flow and motivation or indirectly promote flow via motivation among university students.

More specifically, we examine the possible mediating effect of motivation between the restorative quality perceived in the learning environment and flow.

In this research, we also integrate the JD-R model with the attention restoration theory. ART describes the role of the social–physical environment in reducing stress and allowing cognitive recovery from attentional fatigue. Thus, the environment acts as a resource. In the following, we first explain the construct of restorative quality in environments. In the subsequent sections, we then elaborate on the JD-R model, the construct of motivation, and how motivation mediates the relationship between restorativeness and flow. Lastly, we introduce the hypotheses of the present research. Figure 1 presents the model proposed.

## 2. Theoretical Background and Hypotheses Development

### 2.1. The Perceived Restorative Quality of the Environment in the Academic Context

In the learning environment, a variety of resources or demands may have an influence on individuals. For instance, the perceived quality of the physical environment may determine satisfaction [33,34] and learning efficiency [35,36,37].

Precisely, many aspects of physical design, such as spatial layout or noise, can hinder or improve performance by affecting a student’s physical and psychological resources [37]. More precisely, the environment can allow individuals to relax and distance themselves from everyday thoughts and demands. In this regard, university students have multiple demands placed on them, such as taking exams and engaging in many activities. As a result, they may experience mental fatigue [22,38,39] that, in turn, may reduce their effort level, affect their concentration, and lead to lower academic performance [40]. In this regard, the concept of restorativeness [18] refers to the capacity of the environment “to offer a concrete and available means of reducing suffering and enhancing effectiveness” [41].

Research has paid relatively little attention to the characteristics of learning environments that help students complete restoration to improve their performance.

According to attention restoration theory (ART) [22,23], direct attention is voluntary; it plays a crucial role in controlling distraction, requires effort, and is related to attentional fatigue [41]. The theory describes how the socio-physical environment can support psychological restoration and explains how mental fatigue and direct attention can be restored through four proprieties: Fascination, being away, extent, and compatibility [42].

Fascination is described as an effortless form of attention that allows a fatigued attentional system to rest [41]. This property is present when individuals find a place or situation interesting for them. Being away refers to distancing oneself from routine activities and demands that lead to attentional fatigue. In this condition, students have a sense of being in a different place and/or engaged in different cognitive content [41].

Extent refers to the scope and coherence of the environment that has vast content to the extent that it is possible to get lost in it. Hence, the environment is perceived as a “whole other world” [22]. Finally, compatibility refers to the fit between the demands of the setting and environment and an individual’s goals; the setting and environment should support the actions needed by individuals to achieve their purposes [43].

ART has generally been applied to explain psychological restoration and as a strategy to cope with stress using the natural environment and the learning environment with natural elements [13,44], but recently, some researchers have also examined the role of restoration in the workplace [45,46,47] and in the academic context [48]. In academic environments, Yusli and colleagues [48] found a positive relationship between restorativeness and well-being in a sample of university students.

ART constructs explain how the restorative experience may help students restore or gain internal resources to meet environmental and learning demands.

Therefore, it is important to verify whether the characteristics of the learning environment in reducing stress can be helpful in improving further resources such as, in this study, motivation and flow in the academic context. The relationship between resources and demands and the process that fosters positive outcomes can be described by the JD-R model as follows.

### 2.2. The Job Demands–Resources Model (JD-R)

The job demands–resources model (JD-R) [20] is a conceptual framework used to explain the dynamics of resource depletion and restoration on job or study characteristics. Therefore, this model is relevant in understanding the role of restorativeness in the learning environment.

According to the JD-R model, every job (including student activities) is characterized by job demands and job resources [49,50]. Demerouti and colleagues [49] defined job demands as “all physical, psychological or social aspects of the job that require sustained physical or mental effort and that are therefore associated with psychological costs, such as emotional exhaustion” [14], (p. 501). Examples of job demands (in the learning context) are time pressure, long studying hours, noise, and all elements that drain energy. In university or learning contexts, the number of courses or the number of study hours can contribute to mental demands. In contrast, job resources are defined as “those aspects of the job that are functional in achieving work goals in stimulating personal growth and development, and reducing job demands and the associated psychological costs” [14], (p. 501).

Examples of job resources (in the learning context) include support from teachers, colleagues, and the environment (which helps to achieve an individual’s goals) and performance feedback, which may enhance learning. Job demands and job resources can be both external (e.g., rewards, task variety, and social support) and internal (cognitive) [49]. In the learning context, the JD-R model premises that the combination of high job demands and high job resources results in learning engagement [51].

The JD-R model also incorporates personal resources [52], referring to all aspects of the self that are generally linked to resilience and reflect an enhanced self-perceived ability to successfully influence one’s environment [50,53].

Personal resources positively affect job resources [54] and strengthen the positive relationship between job resources and well-being [54]. Specifically, personal resources are relevant antecedents of motivation and can promote job/academic resources, which, in turn, can further increase personal resources [21].

Restorative environments, or restorativeness, can be considered job resources because of their ability to replenish psychological resources and help students to gain some psychological distance from ordinary activities and engage effortless attention in some interesting activities.

Essentially, the JD-R model combines two psychological processes, a stressful process and a motivational process, which can explain the dynamics of resource depletion and restoration.

A stressful process, due to excessive job demands and lack of resources, may lead to negative outcomes such as poor performance [54,55]. Excessive job demands drain energy and other resources [20]. This stress process is also aligned with the Conservation of Resources (COR) theory [56], which suggests that stress occurs when an individual’s energy resources are depleted or new resources are not available.

A motivational process that is promoted by abundant job resources may lead to positive outcomes such as superior performance [57]. Job resources enhance employee energy and motivation. More precisely, the availability of resources can counteract the negative effects of demands [20,58,59], foster worker growth, learning, and development [60,61], and decrease work stress and burnout in the learning context [62].

Increasing resources protects workers against the adverse effects of job demands and promotes work engagement, whereas a lack of resources could have health-impairing consequences [63]. A recent meta-analysis [64] summarized the positive effect of job resources on work engagement and satisfaction. Generally, resources can positively affect individuals, facilitate their engagement, protect them from psychological discomfort [65], and predict motivation [20].

### 2.3. Restorativeness and Motivation

Based on the JD-R model, which assumes job resources (i.e., restorativeness) enhance employee energy and motivation fostering worker growth, learning, and development [60,61], through a motivational process, we expect that restorativeness is positively linked with student motivation. Motivation in the JD-R model is a mediator of the relationship between job resources (in this study, restorativeness in the learning environment) and positive outcomes (e.g., flow in this case). Generally, the positive effects of the environment have been demonstrated in previous studies [11,47,66,67], but the relationship with restorativeness, motivation, and flow has yet to be considered.

In the following two subsections, we address the relationships between learning environments and two relevant psychosocial dimensions related to learning (i.e., flow and motivation).

### 2.4. Flow

Csikszentmihalyi described the state of flow as ‘‘a sense that one’s skills are adequate to cope with the challenges at hand, a goal-directed, rule-bound action system that provides clear clues as to how well one is performing…concentration is intense…and the sense of time becomes distorted’’ [8]. Therefore, when individuals enter a flow state, distractions are reduced. Flow occurs when an individual’s skills are sufficient to meet the challenges [8] and whenever their skills fit the situational demands [68]. Individuals perceive a challenge–skills balance, and they believe the task is achievable. If the challenge level or demands exceed an individual’s skill or resources for a task, the situation can produce stress, and the individual may disengage. The European Flow-Researchers’ Network [69] defined flow as “a gratifying state of deep involvement and absorption that individuals report when facing a challenging activity and they perceive adequate abilities to cope with it”.

Three conditions are needed to be in a flow state: Clear goals throughout the activity or process, immediate feedback, and a balance between challenges and skills [70].

Flow is positively related to focused attention, losing track of time, being in control, becoming less self-conscious, enjoying what one is doing, and performance [71]. Specifically, in relation to learning aims, flow was found to be positively related to exam performance [72], goal progress [73], and academic success [74,75,76]. It is a form of psychological well-being that is desirable in academic learning contexts [77]. Bakker [78] applied the flow experience to the working condition, comparing the flow state with work engagement. Specifically, he defined flow as a short-term peak experience characterized by absorption (immersion and total concentration in the work), work enjoyment (pleasure experienced by people during work), and intrinsic work motivation (working to feel pleasure and satisfaction).

### 2.5. Restorativeness and Flow

Because the experience of flow, as we have noted, is a balance between skills (or resources) and challenges (or demands), it can be examined according to the JD-R model. Specifically, students can experience a state of flow in the learning context [79] when they can access job resources such as environmental resources in the learning place or when job demands are balanced with high resources (in this case, environmental resources and motivation).

Some authors have found that job resources are an antecedent of flow [80,81] and well-being [78,81].

Specifically, the restorativeness quality of the learning environment, which is functional in achieving work goals and encourages personal growth, development, and learning [60], acts as a job resource, restores direct attention, and promotes concentration through ART.

A recent meta-analysis [82] confirmed that flow had a positive association with many motivational indicators, such as volition, engagement, goal orientation, achievement motive, interest, and intrinsic motivation, and with emotional aspects and performance (because individuals are highly concentrated). Thus, we also expect a positive effect of motivation on flow.

### 2.6. Motivation and Flow

Motivation refers to acting to do or obtain something and may significantly affect higher academic performance [83]. Motivation is an important part of human behavior that influences student energy, persistence in tasks [84], and academic achievement [85,86]. There are two types of motivation, intrinsic and extrinsic [87]. Intrinsic motivation refers to activities carried out for one’s own interest and enjoyment [88]. It refers to activities that provide an individual with personal satisfaction and are not dependent on external rewards [89].

Intrinsic motivation is associated with higher performance, school achievement [90], engagement [91,92,93], and learning and development [94]. Csikszentmihalyi [8] suggests that intrinsic motivation has a relevant role in experiencing a state of flow (which originates from motivation theory) because it serves to energize, direct, and sustain behaviors [95]. Various studies have shown that intrinsic motivation is positively associated with flow, and motivation facilitates flow states [68,96,97]. Therefore, a higher level of individual motivation can, in turn, become an intrinsically enjoyable mental state [5,6] characterized by absorption and intrinsic work motivation defined by Csikszentmihalyi [7,8] as flow.

Conversely, extrinsic motivation depends on external factors. For example, individuals are motivated by rewards, including in the form of social approval or appreciation. Nevertheless, even when an individual is not intrinsically motivated, extrinsic motivation can positively affect well-being, performance, and outcomes when it is generated by values with which the person identifies [87]. Generally, motivated individuals are more likely to experience flow [96,98]. In addition, motivated students are more likely to engage in a learning context and experience more flow than less motivated students [99]. In their empirical research, Csikszentmihalyi and Nakamura [9] observed that when people show an interest in the activity, they can be absorbed with high levels of engagement and concentration.

Recently, Kong and Wang [100] found a positive relationship between the perception and support of parents and students’ flow experience through the mediating role of student learning motivation.

Therefore, motivation is a relevant resource to promote positive outcomes such as flow. It is relevant to note that resources and flow mutually influence each other: Resources can predict flow and flow leads to a greater perception of job resources in a virtuous circle [80].

By contrast, a lack of resources has a detrimental effect on worker motivation and performance because it impedes the achievement of goals and the possibility of learning [101], as reported in the JD-R model.

### 2.7. Study Aims and Hypotheses

The present study aims to explore the positive relationship between the perceived restorative quality of settings available for student learning and flow via intrinsic and extrinsic motivation. Based on the literature we have cited and the JD-R model and ART, the following hypotheses are proposed:

**Hypothesis** **1** **(H1).**
*The restorativeness of the learning setting (compatibility, extent, being away, and fascination) is positively associated with the flow state.*


**Hypothesis** **2a** **(H2a).**
*Intrinsic motivation mediates the relationship between the restorativeness of the learning setting (compatibility, extent, being away, and fascination) and flow state.*


**Hypothesis** **2b** **(H2b).**
*Extrinsic motivation mediates the relationship between the restorativeness of the learning setting (compatibility, extent, being away, and fascination) and flow state.*


## 3. Methods

### 3.1. Procedure and Participants

The study was conducted in February 2022 at two Italian universities. Participants were students in the 3-year psychology and health courses. They were enrolled in the 2021–2022 academic year. Students were selected based on convenience and accessibility. They came from seven psychology classes. The overall sampling method was purposeful to obtain homogeneity in the two samples, with the aim of preliminary testing of the theoretical framework in the Italian context. Specifically, the two Italian samples and contexts are homogeneous for age, year of course, and characteristics of the environment and place where the students studied.

Further, we also considered the values of Kaiser–Meyer–Olkin, and the results of the Bartlett test of sphericity to understand the adequacy of the sample. Values of KMO were higher than 0.790, and the Bartlett test was significant, supporting the adequacy of our sample. Students were invited to participate in this research project by a teacher/researcher during the lessons. The teacher provided information about the aim of the study and what taking part in it involved.

Participants responded to an online questionnaire, which provided a range of information explaining the aim of the study, the anonymity of data, and the voluntary basis of participation. They filled out a questionnaire at home or at the university campus.

Overall, 231 students were informed of the possibility of participating in this study during the lesson, but only a total of 165 of them completed the questionnaire. None were incomplete. Regarding gender, 26.7% of students were men, 73.3% were women, and most (73.3%) were in the age range of 19 to 24 years. First-year students accounted for 38.8%, 18.8% of the sample were in their second year, 29.1% were in their third year, 4.2% were in their fourth year, and the remaining 5.5% were outside the prescribed time. Students who said they mainly studied at home accounted for 51.5%, while 48.5 % of students said they mainly studied at the university (classroom, reading rooms, or library). An a priori power analysis was performed to calculate an adequate sample size [102] for a linear multiple regression model. To perform the power analysis test, a large effect size (i.e., 0.8), an alpha of 0.05, and a power of 0.95—with two predictors—were input. The results of our analysis determined that a minimum sample size of at least 50 students was required.

### 3.2. Measures

The study considered three measures categorized on the basis of the literature. Specifically, the study identified perceived restorativeness, motivation, and flow.

Perceived restorativeness was made up of 16 items from an Italian version [103] of the Perceived Restorativeness Scale (PRS) [18]. Instructions for the scale directed students in the learning environment. Each item was rated on a 7-point scale, from strongly disagree (=0) to strongly agree (=6). Examples of items include: “Spending time here gives me a good break from my day-to-day routine” (being away); “I would like to spend more time looking at the surroundings” (fascination); “I have a sense of oneness with the setting” (compatibility); and “There is too much going on” (coherence as an aspect of extent; negatively formulated).

Motivation was measured with eight items of the Twelve Dimensions of the College Competence Scale [104]. Respondents rated their level of agreement with the survey statements on a 5-point scale with options ranging from not at all (=1) to completely (=5). Examples of items include: “Each course I attend teaches me something I am interested in” for intrinsic motivation; “Even the days when I feel a bit lazy, I can find a way to study at least a bit” for extrinsic motivation.

A flow scale [70] made up of 8 items, from an Italian translation [105], was used in the present study. The response format of the scale was a 7-point Likert scale ranging from strongly disagree (=0) to strongly agree (=6). Instructions for the scale were focused on students’ activities. A sample item from the scale is “I have a high level of concentration”.

### 3.3. Socio-Demographic and Learning Context Control Variables

The questionnaire also included questions about socio-demographic characteristics and learning context variables. Those used in the present analyses were gender (men = 1, women = 2), age (continuous variable), type of university attended by students (two categories), years of study (six categories, from first year = 1 to sixth year = 6), and place of study (two categories, 1 = university; 2 = home).

### 3.4. Data Analysis

We performed CFAs (confirmatory factor analyses) to evaluate the factorial structure of each of the three measures used in the study. The goodness of fit was evaluated using the chi-square value and fit indices satisfying standard criteria (e.g., Comparative Fit Index > 0.90 [106], Tucker–Lewis index (TLI) [107], Root Mean Square Error of Approximation index < 0.08 (RMSEA) [108], and Standardized Root Mean Square Residual index (SRMR) [109].

Further, we examined the potential effects of common method bias (CMB) comparing two different models using Harman’s single-factor procedure [110], which was tested using the AMOS statistical package in SPSS version 20. First, exploratory factor analysis was conducted to determine the number of factors necessary to account for the variance in the variables. Bias is indicated when a single factor explains a majority of the total variance. Second, a confirmatory factor analysis, considering a model with three latent variables, was performed and compared with a one-factor model. If common method variance is largely responsible for the associations among the variables, the one-factor CFA model (the simplest model) should fit the data well [111]. SPSS 20 was also used to calculate means, standard deviations, and alpha reliabilities (α) for each scale and correlations (Pearson’s r) between variables, and to test hypotheses H1, H2a, and H2b.

Lastly, to test hypothesis H1, we evaluated the effect of restorativeness sub-scales for predicting flow in the hierarchical multiple linear regression while controlling several socio-demographic and learning context variables. Two models were tested. In the first model/block, flow and intrinsic and extrinsic motivation were separately regressed with socio-demographics and learning context characteristics and then analyzed to check for a possible confounding effect. In the second block (model 2), motivation and flow were regressed with each restorativeness sub-scale. Socio-demographics and learning context characteristics significantly associated with flow were included as covariates in all subsequent analyses to ensure that the association between the restorativeness sub-scales and flow was not spurious. Further, the independent sample Mann–Whitney U test for samples that are not normally distributed was conducted on different two samples for significant socio-demographics and learning context characteristics that had different patterns in flow and in intrinsic and extrinsic motivation. Furthermore, we split the sample into groups based on the results of the two-sample test and then conducted regression analyses for each group separately, using the same procedures for the mediation described in the following. Mediation was performed by directly testing the significance of the direct effect of each sub-dimension of restorativeness on flow and through the mediator (hypotheses H2a and H2b). To estimate the significance of the indirect effects, we used the bootstrapping approach with 5000 samples [112], which is based on a mean derived from n samples with replacement. The analysis was performed using the SPSS statistical package and PROCESS script version 4.1 [113]. Restorativeness sub-scales were entered in the PROCESS script as independent variables, intrinsic and extrinsic motivation as mediator variables, flow as the dependent variable, and significant socio-demographics and learning context variables as covariates.

## 4. Results

### 4.1. Descriptive Statistics

Bivariate Pearson correlations among the measured variables considered in the study were calculated and are presented in Table 1. This table also reports means and standard deviations. For restorativeness, we found that sub-scale scores of fascination, being away, and compatibility were significantly positively correlated with extrinsic motivation and flow, whereas extent was significantly negatively correlated with intrinsic motivation but did not correlate significantly with each other sub-dimension of restorativeness quality of the learning context, extrinsic motivation, and flow. Among the several control variables, only gender was significantly negatively related to fascination. Moreover, fascination had a positive correlation with the place where students work. Finally, the type of university attended by students had a significant negative correlation with compatibility.

### 4.2. Results

#### Confirmatory Factor Analyses and Reliability

All the assessed measures showed an acceptable model fit. Furthermore, all the variables showed good reliability with Cronbach’s alphas ranging from 0.64 to 0.91, as reported in the following.

Perceived Restorativeness Scale. CFA supported the four-factor structure (χ^2^ = 62.578, df = 29, *p* = 0.000, χ^2^/df = 2.158, CFI = 0.930, TLI = 0.892, SRMR = 0.059, RMSEA = 0.084). Cronbach’s alpha was 0.68 for fascination, 0.77 for compatibility, 0.64 for extent, and 0.83 for being away.

Motivation (intrinsic and extrinsic). CFA supported the two-factor structure (χ^2^ = 12.964, df = 8, *p* = 0.113, χ^2^/df = 1.621, CFI = 0.992, TLI = 0.1, SRMR = 0.027, RMSEA = 0.062). Cronbach’s Alpha was 0.87 for extrinsic motivation and 0.89 for intrinsic motivation.

Flow scale. CFA supported the one-factor structure (χ^2^ = 17.443, df = 12, *p* = 0.134, χ^2^/df = 1.454, CFI = 0.994, TLI = 0.987, SRMR = 0.026, RMSEA = 0.053). Cronbach’s alpha = 0.91.

### 4.3. Evaluation of Common Method Bias

We evaluated the potential effect of the common method bias. Exploratory factor analysis (EFA) was performed with different methods (unrotated principal components factor analysis, principal component analysis with varimax rotation, and principal axis analysis with varimax rotation). EFA conducted with different methods showed the presence of six factors. Together, these six factors accounted for 70.6% of the total variance. No dominant factor accounted for more the 50% of the variance; the first (largest) factors accounted for 33.4% of the total variance. Moreover, two different models were compared using a confirmatory factor analysis (CFA) considering the three constructs used in this study (restorativeness, motivation, and flow) with regard to a one-factor model with all items loading on one factor. We found that the single-factor (common method) model did not fit the data well (χ^2^ = 1609.309, df = 277, *p* = 0.000, χ^2^/df = 5.810, CFI = 0.951, TLI = 0.382, SRMR = 0.198, RMSEA = 0.171) compared with one model with three constructs (χ^2^ = 319.896, df = 222, *p* = 0.000, χ^2^/df = 1.441, CFI = 0.955, TLI = 0.945, SRMR = 0.055, RMSEA = 0.052). Thus, indices support the relationships between distinct latent variables. Furthermore, a chi-square comparison is significant (chi-square difference = 1289.413 with 55 df; *p* < 0.001); thus, we can state that CMB is not a substantial concern in this study.

### 4.4. Hypothesis Tests

As shown in Table 2, we found that gender had a positive effect on flow (β = −0.181, *p* < 0.05), but age, type of university, and place where students study did not contribute significantly to explaining intrinsic and extrinsic motivation or flow in the linear hierarchical regression analysis (first step, model 1). In the first step, model 1, with socio-demographic and learning context control variables and intrinsic and extrinsic motivation regarding flow, did not show an acceptable fit (*F* > 0.05). The results of the regression analysis in the second step confirmed a negative effect of gender on flow (*β* = −0.163, *p* < 0.05) and on intrinsic motivation (*β* = −0.08 *p* < 0.005).

The Mann–Whitney U test showed differences between males (M = 5.07, SD = 0.91) and females (M = 4.53, SD = 0.13) regarding flow (z = −2.81, *p* <0.05) but not between males (M = 3.51, SD = 0.91) and females regarding intrinsic motivation (M = 3.29, SD = 0.92, z = −1.56, *p* > 0.05) or between males (M = 3.71, SD = 0.91) and females regarding extrinsic motivation (M = 3.60, SD = 0.92, z = −0.67, *p* > 0.05). To simplify Table 2, we only reported the relationship between socio-demographic and learning context control variables and flow for model 2, but we pared intrinsic and extrinsic motivation (mediators) from model 2.

The results confirmed a positive effect of compatibility (*β* = 0.416, *p* < 0.001), being away (*β* = 0.330, *p* < 0.001), and fascination (*β* = 0.253, *p* < 0.01) on flow but extent did not have a significant relationship with flow (*p* < 0.05), as reported in Table 2. Hypothesis 1 was partially supported.

In regard to the mediating role of intrinsic and extrinsic motivation on the relationship between the restorativeness sub-scales of the learning setting and flow, we found an indirect effect of compatibility on flow through intrinsic motivation (*β* = 0.114, *p* < 0.05; 95% CI = 0.328 to 0.198) and extrinsic motivation (*β* = 0.139, *p* < 0.05; 95% CI = 0.054 to 0.217); extent on flow through intrinsic motivation (*β* = 0.095, *p* < 0.05; 95% CI = 0.016 to 0.179); being away on flow through extrinsic motivation (*β* = 0.087, *p* < 0.05; 95% CI = 0.019 to 0.159); and fascination on flow through extrinsic motivation (*β* = 0.092, *p* < 0.05; 95% CI = 0.019 to 0.168).

However, the results did not confirm a significant indirect effect of extent on flow through extrinsic motivation, being away on flow through intrinsic motivation, or fascination on flow through intrinsic motivation. Thus, the results partially support hypotheses H2a and H2b.

Generally, these results showed that intrinsic and extrinsic motivation act as mediators in the relationship between restorativeness and flow except for the relationship between extent and flow through extrinsic motivation, being away on flow through intrinsic motivation, and fascination on flow through intrinsic motivation.

Furthermore, as reported in Table 3, the direct relationships of the restorativeness sub-scales (in the presence of the mediators) with flow were also significant for compatibility with the presence of intrinsic motivation (*β* = 0.279, *p* < 0.01; 95% CI = 0.148 to 0.384) and extrinsic motivation (*β* = 0.261, *p* < 0.01; 95% CI = 0.126 to 0.373), for being away with the presence of intrinsic motivation (*β* = 0.261, *p* < 0.001; 95% CI = 0.097 to 0.266) and extrinsic motivation (*β* = 0.239, *p* < 0.01; 95% CI = 0.079 to 0.254), and for fascination with the presence of intrinsic motivation (*β* = 0.179, *p* < 0.05; 95% CI = 0.044 to 0.250) and extrinsic motivation (*β* = 0.160, *p* < 0.05; 95% CI = 0.025 to 0.237). Although the beta coefficients of these independent variables on flow were significant, they were smaller than those without the presence of the mediator, as shown in Table 3, implying a mediating role of motivation.

However, the direct relationship between extent and flow was not significant with the presence of the mediator both for extrinsic (*p* > 0.05) and intrinsic motivation (*p* > 0.05), as reported in Table 3.

Finally, the simple relationship (total effects) between restorativeness sub-scales and flow was positive and significant for compatibility (*β* = 0.393, *p* < 0.05; 95% CI = 0.242 to 0.509), being away (*β* = 0.323, *p* < 0.05; 95% CI = 0.127 to 0.328), and fascination (*β* = 0.252, *p* < 0.05; 95% CI = 0.086 to 0.328), but the simple relationship between extent and flow was not significant (*p* > 0.05).

Thus, the results support the full mediation effect of extent on flow through intrinsic motivation and a partial mediation effect of compatibility on flow both through intrinsic and extrinsic motivation, of being away on flow through extrinsic motivation, and of fascination on flow through extrinsic motivation.

Finally, the results for the female group (n = 121) resemble that of the full sample. The only difference we found for the female group was a non-significant mediation effect of being away on flow through extrinsic motivation (*β* = 0.526, *p* > 0.05; 95% CI = −0.043 to 0.117). However, the results for the male group (n = 44) were considerably different compared with the full sample. Notably, the results confirmed just two significant indirect effects, namely, compatibility on flow through extrinsic motivation (*β* = 0.153, *p* < 0.05; 95% CI = 0.150 to 0.315) and being away on flow through extrinsic motivation (*β* = 0.861, *p* < 0.05; 95% CI = 0.004 to 0.210). Moreover, for the male group, regarding the simple relationship between restorativeness sub-scales and flow, only compatibility had a significant association with flow (*β* = 0.293, *p* < 0.05; 95% CI = 0.096 to 0.490). In other words, for the male sub-sample, extrinsic motivation acts as a partial mediator between compatibility and flow and as a full mediator between being away and flow.

## 5. Discussion

In the present study, we aimed to address the role of perceived quality of the academic learning context referring to four environmental proprieties (compatibility, being away, extent, and fascination) in promoting flow, both directly and indirectly through the mediational effect of motivation (intrinsic and extrinsic) on flow. An innovative aspect of this study was to illustrate the role of the four properties of the environment in the academic context, extending the use of the JD-R based on the assumption that restorativeness may be considered a job resource because it helps achieve work and learning goals and reduces learning and work context demands [49]. Overall, consistent with previous studies [11,13], the results supported our hypotheses both for the main effects of restorativeness and the mediational effect of motivation [51,94,100], revealing that compatibility, being away, and fascination had a positive association with flow [80,81] (H1). However, we did not find such an association between extent and flow. Furthermore, our results confirmed that extrinsic motivation acted as a partial mediator (H2a) through the relationships between compatibility and flow, being away and flow, and fascination and flow, whereas intrinsic motivation (H2b) revealed a partial mediating effect through the relationships between compatibility and flow, and its full mediating role through extent and flow. The results did not support the mediating role of intrinsic motivation between being away and flow or fascination and flow. Our results also indicated that extrinsic motivation was not a mediator between extent and flow.

Generally, these results are aligned with the JD-R model [20,21,64] and the findings of previous studies [45,46,47,48] that have pointed out the capacity of job resources to promote positive outcomes, such as motivation [20,96], engagement [11,64], and flow [80,81]. Notably, our findings underline that three of the four restorativeness properties of the learning environment (compatibility, being away, and fascination)—which are considered job resources—foster students’ learning goals [60,61] through a motivational process. These findings align with previous research, which supports the positive effect of resources on students’ psychological discomfort [65]. These results are important in the learning context where the four properties of the perceived restorativeness quality play a relevant role for students in improving their concentration (i.e., flow) and academic success [40]. Specifically, students may experience a state of flow [79,82]—which is characterized by a balance between skills (or resources) and challenges (or demands)—when they have access to environmental resources at their place of learning. As shown by previous studies [11,13], the restorativeness properties of the environment or high resources balance the job demands of the learning context allowing psychological restoration of those resources that are depleted when individuals work and study. Such results highlight that the availability of environmental job resources may lead to positive outcomes such as superior performance [57] and counteract the negative effects of demands [20,58,59], decreasing work stress in the learning context [62].

These results also align with the attention restoration theory (ART) in relation to the replenishment of internal resources, which is needed to meet learning demands and promote success in the academic context [48]. Conceptually, the learning and socio-physical environment could provide students with more resources for learning in terms of either motivation and/or flow, supporting them through the psychological restoration process ensured by the four proprieties of the environment. However, in this study, it is important to note the role of extent as the only of the four restorativeness properties that did not have a main effect on flow. Although these results do not support our hypotheses, they align with earlier studies that showed that when campus settings had insufficient restorative objects, they did not induce significant mental restoration [114]. Students cannot be fully immersed in such settings because of demands such as exams, assignments, and responsibilities and shift attention away from their campus life, daily activities, and stressful feelings [115].

Looking at items of extent (e.g., there is too much going on it, it is a confusing place), they clearly refer to a form of distraction. According to the ART, extent should help to replenish resources, but it showed a significant negative association with compatibility (i.e., fit between demand setting and environment and an individual’s goals) (r = −169) at a high level in this study (M = 5.28 on a scale from 1 to 7). Presumably, such a high level of extent reflects some form of excessive and negative distraction that does not support students’ goals.

In contrast to the main effects, extent was significantly (and positively) associated with intrinsic motivation. Here, it is interesting to consider why extent promoted intrinsic motivation. We hypothesized that while students try to meet the high environmental demands (deducible by the high levels of extent) according to the JD-R model and the ART, they are forced to find new resources, especially intrinsic motivation, to resolve this negative situation.

There are further important questions regarding the different and salient roles of compatibility in promoting both intrinsic and extrinsic motivation, and of being away and fascination and repeating the same positive pattern exclusively for extrinsic motivation rather than for intrinsic motivation.

Numerous factors may be at play (individuals, groups, organizational, and situational). Specifically, in this study, we detected a higher level of compatibility (M = 4.68) compared with the other sub-dimensions of restorativeness for both fascination (M = 3.74) and being away (M = 3.37). Compatibility, as noted above, was needed for students to adapt to their learning environment to meet demands and achieve their goals, but when demands increased and it was not possible for them to adapt their environment, they were forced to find new (internal) resources, such as intrinsic motivation, that in turn could promote flow [97]. An alternative explanation could be associated with the meaning of being away and fascination. Being away, referring to gaining some psychological distance from the learning context [41], and fascination, engaging effortless attention in some pleasant activity, seem to act as a form of distraction in the learning context [41], similar to extent, but in a positive way. Thus, being away and fascination did not have the same capacity for compatibility in promoting internal resources.

As relatively few researchers have addressed the specific problem in focus here, there needs to be a wealth of recent literature to link our findings with recent results. Thus, our interpretation needs further testing in future studies.

Looking at the overall results, the present study extends the existing literature by confirming that the restorativeness properties of the academic context environment can be significant resources to improve motivation and flow. Our findings suggest that students should choose or adapt their environment to promote positive outcomes (such as motivation and flow) but also further resources. However, some properties of the environment should be controlled because they can become forms of distraction. In some cases, some properties of the environment may be positive forms of distraction (i.e., being away, fascination) and, in other cases (i.e., extent), a negative form of distraction. The significant effects of restorativeness on flow and motivation have implications for the design of the learning context since they demonstrate the need to pay attention to the physical environment and, specifically, to compatibility dimensions that act via flow, but also both via intrinsic and extrinsic motivation. Regarding this last point, the results also suggest that researchers should pay attention to high levels of the four properties (especially for extent and compatibility), and students, teachers, designers, and workers could measure the levels of each property and compare them with prior defined standards that ensure intrinsic motivation and flow.

Finally, we also computed additional analyses for both male and female sub-samples. Generally, our results aligned with our hypotheses for the female sample and the full sample but not for the male sample. We noted for men that only compatibility had a significant effect on flow, and extrinsic motivation acted as a partial mediator between compatibility and flow and as a full mediator between being away and flow.

These results are relevant to deal with the different environmental perceptions of men and women and for understanding the possible effect of restorativeness on learning outcomes.

These results further highlight the importance of differentiating learning environments based on gender.

### 5.1. Limitations of the Research

Although our results support most of our hypotheses, we used a cross-sectional correlational design that is relatively sufficient to demonstrate the causal relationship between variables. Furthermore, even if we used socio-demographic and learning context variables to control possible confounding effects, the participants’ samples were statistically small, so generalizations are tricky. The limited sample size can only provide preliminary indications and does not allow for generalizing the results of Italian students. Finally, we selected the sample based on convenience and accessibility, so the sample is not representative. A larger sample would help to examine the interaction effect of multiple variables in a more complex model. Nevertheless, further research should address the detected limitations and study different student groups and academic learning contexts to generalize the findings. Finally, the sample consisted mainly of women (73.3%) compared to men (26.7%) and, therefore, was mainly biased when comparing the two subsamples and confirming the difference between males and females.

### 5.2. Conclusions

Our findings suggest that restorativeness plays a role in promoting motivation and flow in the learning context. Further, they shed some light on the role of extent and compatibility in this positive process. These results point to a promising direction for future research and offer practical ideas for the learning context. For instance, future research could focus on specific features of the learning environment able to increase positive outcomes. Designers, students, and teachers could be provided with new tools that support learning and well-being by improving “compatibility” between demands and individuals’ goals and controlling levels of extent, being away, and fascination. It would also be interesting for further research to explore the role of gender in the relationship between restorativeness and positive outcomes and use longitudinal studies to investigate fluctuations in restorativeness and flow over time. In addition, studies could consider further mediators and moderators between restorativeness (and each sub-dimension with different levels) and flow or other important outcomes related to learning performance.

More generally, the results may develop into a new pathway for learning and work research to examine the role of socio-physical and psychosocial aspects of environments on related outcomes.

## Figures and Tables

**Figure 1 ijerph-19-15263-f001:**
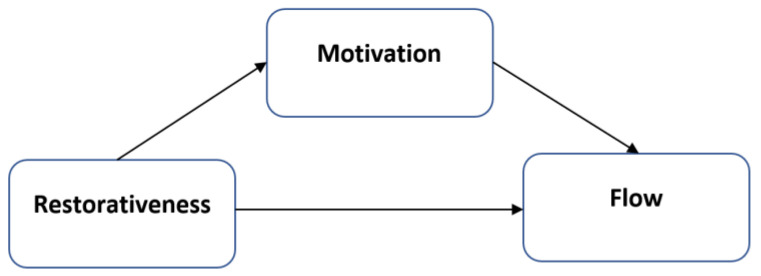
Conceptual mediational model.

**Table 1 ijerph-19-15263-t001:** Study Variables: Descriptive Statistics and Bivariate Correlations (N = 165).

	*M*	*SD*	1	2	3	4	5	6	7	8	9	10	11	12	
Gender	-	-	1												
2. Age	23.4	5.79	−0.024	1											
3. Univers.	-	-	−0.050	0.120	1										
4. Year of study	-	-	0.113	0.095	0.496 **	1									
5. Place of study	-	-	−0.046	0.004	0.022	0.100	1								
6. Compati.	4.68	1.32	0.000	−0.117	−0.190*	−0.141	0.039	1							
7. Being A.	3.37	1.81	−0.030	0.024	−0.080	0.037	0.043	0.435 **	1						
8. Extent	5.28	1.37	−0.057	0.017	0.110	0.093	0.028	−0.169 **	008	1					
9. Fasc.	3.74	1.54	−0.024	0.064	−0.055	−0.032	0.027 **	0.429 **	0.318 **	−0.303 **	1				
10. Flow	4.68	1.26	−0.187 *	0.016	0.034	0.006	0.115	0.394 **	0.333 **	0.078	0.257 **	1			
11. Intrinsic Mot	3.35	0.925	−0.107	−0.041	−0.002	−0.058	0.047	0.237 **	0.131	0.178 *	0.142	0.564 **	1		
12. Extrinsic Mot.	3.64	0.920	−0.050	−0.064	−0.050	−0.086	0.072	0.295 **	0.183 **	0.053	0.188 *	0.532 **	0.716 **	1	

Note: For sex (two categories), universities (two categories), year of study (six categories), and place of study, means and standard deviations were not reported because these variables were categorical in the questionnaire. Sub-dimensions of restorativeness were measured on a 7-point scale, with higher values indicating greater perceived restorativeness. Flow was measured on a 7-point scale, with higher values indicating higher levels of the construct. Intrinsic and Extrinsic motivation were all measured on 5-point scales with higher values indicating higher levels of the constructs. * *p* < 0.05; ** *p* < 0.01.

**Table 2 ijerph-19-15263-t002:** Hierarchical Regression Analyses of the Independent and Interactive Associations of Socio-Demographic and contextual variables, and Sub-Dimension of Perceived Restorativeness (Compatibility, Extent, Being Away, Fascination) with Flow, and Intrinsic and Extrinsic Motivation (N = 165).

	Flow			Intrinsic Motivation			Extrinsic Motivation					
Model 1	*β*	*t*	*P*	*β*	*t*	*P*	*β*	*t*	*P*			
Gender	−0.181	−2.305	<0.05	−0.097	−1.220	0.224	−0.039	−0.488	0.627			
Age	0.008	0.109	0.914	−0.041	−0.515	0.607	−0.057	−0.713	0.477			
University	0.019	0.213	0.832	0.028	0.303	0.763	−0.008	−0.085	0.933			
Year of study	0.005	0.056	0.956	−0.061	−0.663	0.508	−0.080	−0.864	0.389			
Place of study	0.106	1.354	0.178	0.048	0.604	0.547	0.078	0.988	0.325			
*R* ^2^	0.047			0.018			0.018					
Adjusted *R*^2^	0.017			−0.013			−0.013					
Omnibus test of the regression	*F*(5, 159) = n.s			F(5, 159) = n.s			F(5, 159) = n.s					
	Flow			Flow			Flow			Flow		
Model 2												
Gender	−0.181	−2.526	<0.05	−0.178	−2.25	0.005	−0.166	−2.230	0.027	−9.176	−2.301	0.023
Age	0.047	0.661	0.510	0.008	0.108	0.026	−0.001	−0.017	0.986	−0.010	−0.128	0.898
University	0.081	0.973	0.332	0.014	0.159	0.914	0.065	0.756	0.451	0.035	0.397	0.692
Year of study	0.032	0.381	0.704	0.001	0.014	0.874	−0.029	−0.341	0.733	0.007	0.082	0.935
Place of study	0.085	1.197	0.233	0.105	1.33	0.989	0.095	1.282	0.202	0.099	1.305	0.194
Compat.	0.416	5.734	<0.001									
Extent				0.063	0.802	0.424						
Being away							0.330	4.463	<0.001			
Fascination										0.253	3.354	<0.001
*R^2^*	0.211			0.017				0.154		0.110		
Adjusted *R^2^*	0.181			0.051				0.121		0.076		
Omnibus test of the regression	F(6, 158) = <0.001			F(6, 158) = n.s				F(6, 158) = <0.001		F(6, 158) = <0.01		

Note: In Model 2, given the significant effect of only gender on intrinsic motivation, to simplify the reading of this table we reported results in the text and pared intrinsic and extrinsic motivation from the table.

**Table 3 ijerph-19-15263-t003:** Direct and Indirect effects of fascination (FA), extent (EX), being away (BE), and compatibility (CO) on flow through intrinsic (IM) and extrinsic motivation (EM), and total effects.

Model	Direct E. Estimate	Direct Effect		Indirect E. Estimate	Indirect E. 95% BC Boostrap CI (5.000 Samples)		Total E. Estimate	Total Effect	
		LLCI	ULCI		LLCI	ULCI		LLCI	ULCI
CO → IM → Flow	0.279	0.148	0.384	0.144	0.328	0.198	0.393	0.242	0.509
CO → EM → Flow	0.261	0.126	0.373	0.139	0.054	0.217			
EX → IM → Flow	−0.282	−0.144	0.093	0.095	0.016	0.179	0.672	−0.784	0.2023
EX → EM → Flow	0.041	−0.814	0.157	0.026	−0.062	0.119			
BA → IM → Flow	0.261	0.097	0.266	0.065	−0.144	0.153	0.323	0.127	0.328
BA → EM → Flow	0.239	0.079	0.254	0.087	0.019	0.159			
FA → IN → Flow	0.179	0.044	0.250	0.073	−0.037	0.152	0.252	0.086	0.328
FA → EX → Flow	0.160	0.025	0.237	0.092	0.019	0.168			

Note: BC = Bias Corrected; CI = Confidence Interval; LLCI = Lower Limit CI; ULCI = Upper Limit CI.

## Data Availability

The data supporting the conclusions of this article will be made available by the authors on reasonable request from the corresponding author.

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
