# Peer review of "The Role of a Restorative Resource in the Academic Context in Improving Intrinsic and Extrinsic Motivation and Flow within the Job Demands–Resources Model"

_ijerph, 2022, doi:10.3390/ijerph192215263_

Round 1
Reviewer 1 Report
Dear Authors,
It is my pleasure reviewing your manuscript on restorative resources and university students’ flow. Although I highly appreciate the theoretical rigor of your work, I noticed serious issues in the methodology and statistical analysis sections that need attention and correction/justification by the authors to claim significant contribution. I hope my comments and suggestions will be taken positively for the improvement of the manuscript.
Best of luck
Summary and General Assessment:
Based on Job Demands Resources (JD-R) model and Attention Restoration Theory (ART), the current study is an effort to add value into the university students’ positive outcome literature in the form of ‘flow’ by theorizing ‘Restorativeness’ (taken as four learning environment factors) and motivation as distal and proximal predictors of flow, respectively. Results of the hierarchical regression analysis and Process macro analysis conducted on a sample of 165 Italian students supported most of the hypothesized relationship of the study.
Overall, the manuscript is well written. The language structure and readability is of excellent quality. The selection of variables in filling the found research gap and to contribute into the relevant body of knowledge is well justified. The theoretical discussions based on JD-R and ART especially provide a solid foundation to the proposed research model of this study. It was easy to follow through the hypotheses development. All the constructs have been acceptably defined and the selection is found justified.
Major Concerns and Recommendations:
The following are my major concerns regarding this research manuscript. Kindly address and respond accordingly.
1. Kindly justify why only two Italian universities were taken for target population of this study? It is important because the limited scope of the study might significantly limit even theoretical generalization of this study. Please provide justifications on your choice.
2. The target population of this study is not clearly explained. Authors need to explain and justify the criteria on which the respondents were targeted. Explain procedures as well please. Furthermore, despite the fact that university students comprise of a large population, why the sample of this study is relatively small (165)? Perform power analysis please to justify your sample size.
3. The authors need to clearly explain and justify the operationalization of “Restorativeness” whether this construct was treated as a multidimensional or unidimensional variable, and why? Understanding the structure of a construct is important in developing and strengthening a concept.
4. Directly related with the point 3, please explain in detail the CFA process. How it was performed? How the variables were treated during CFA? Whether dimensions of the variables or variables were taken while performing CFA? Please explain with justification for your choices.
5. Another serious concern is that why Process macro was needed for this study? As Process macro has significant limitations, especially in the case of this study where simultaneous analysis of all four environmental factors sounds more robust, I did not find any particular justification. Please justify why AMOS was not used in testing mediation hypotheses?
6. The last concern, and the most serious one, is that this study performed EFA and CFA on the same relatively small sample, which is not acceptable. Please see the reference given below for more guidance and justify why both EFA and CFA were needed?
Green, J. P., Tonidandel, S., & Cortina, J. M. (2016). Getting through the gate: Statistical and methodological issues raised in the reviewing process. Organizational Research Methods, 19(3), 402-432.
Author Response
Thank you very much to taking time to read our paper and indicating way to improve it. We appreciate your positive evaluation of our work.
Best regards

Reviewer 2 Report
The abstract It was interesting to read your paper. The topic addressed is relevant and the approach is pertinent, however, I would like to point out some issues here to be revised:
- The abstract is not written well. it can be improved by mentioning design and methodology.
- There are issues in the Introduction setting as the authors should clearly define the theories that are used in the context of their study while integrating theoretical premises.
- There are also issues in the manuscript setting as the literature is not reported as a separate section and the current case remains vague, i.e. it can be addressed in section 2. Theoretical background and hypothesis development.
- The hypothesis is not justified, the above mention can be addressed here, in section 2. theoretical background and hypothesis development, must be included in the justification of the hypothesis. i.e. 2.1. Restorative and Motivation, justify it then hypothesized it.
- The sampling method must be introduced.
- How was the sample size was calculated?
- Possibly a larger sample would be desirable, therefore it is suggested to justify the sample population.
- Discussion should not only interpret the results and findings but further must link to the recent findings in the literature to highlight the contributions. this section needs to be revised.
Author Response
Thank your very much to you for indicating ways to improve our paper.
Best regards.

Round 2
Reviewer 1 Report
Thank you for incorporating / addressing my concerns and comments in the revised version. The revised version of your work is acceptable as a "preliminary" study providing novel directions for future research.
Wish you all the best.
Author Response
Thank you very much for your positive evaluation of our work.
All the best.
Reviewer 2 Report
The author/s has operated a number of changes; however, the issue stands still.
1. The new version of the manuscript should be appropriately revised by applying all comments.
2. The author/s should address the method that has been used to employ the data. (the above comment (1) should be considered here).
3. In section 2 there is still a lack of justifying the hypothesis, i.e. the justification of the relationship between Restrorativeness and Flow/Restorativeness and Motivation therefore the autor/s should address them in sections 2.5 and 2.6
4. Sample size and sampling calculation methods should describe in the methods section. (the above comment (1) should be considered here).
5. In section 3, the author/s jumped from 3.2 to 3.4, therefore there is a missing part that has to be fixed.
6. Justification of the sample size still remains the same in the manuscript. (the above comment (1) should be considered here).
7. The discussion part still remains vague, the author/s should refer to my first-round comment in this regard.
8. It is strongly recommended to have the proposed paper revised by a native English speaker mainly as a phrase topic, i.e. place and order of words.
Author Response
Dear reviewer,
thank you for taking time to read our paper. Please, could you give us more information about your comments number 2. What do you mean by "employ" the data? How the data have been collected? We would really like to improve our work.
Thank you again for you patience.
Best regards
